# Cortex commands the performance of skilled movement

Jian-Zhong Guo, Austin R Graves, Wendy W Guo, Jihong Zheng, Allen Lee, Juan Rodríguez-González, Nuo Li, John J Macklin, James W Phillips, Brett D Mensh, Kristin Branson, Adam W Hantman*

Janelia Research Campus, Howard Hughes Medical Institute, Ashburn, United States

**Abstract** Mammalian cerebral cortex is accepted as being critical for voluntary motor control, but what functions depend on cortex is still unclear. Here we used rapid, reversible optogenetic inhibition to test the role of cortex during a head-fixed task in which mice reach, grab, and eat a food pellet. Sudden cortical inhibition blocked initiation or froze execution of this skilled prehension behavior, but left untrained forelimb movements unaffected. Unexpectedly, kinematically normal prehension occurred immediately after cortical inhibition, even during rest periods lacking cue and pellet. This 'rebound' prehension was only evoked in trained and food-deprived animals, suggesting that a motivation-gated motor engram sufficient to evoke prehension is activated at inhibition's end. These results demonstrate the necessity and sufficiency of cortical activity for enacting a learned skill.

*For correspondence: hantmana@janelia.hhmi.org

**Competing interests:** The authors declare that no competing interests exist.

## Introduction

Execution of complex voluntary movements depends on many functions including choosing a behavior, specifying each step, enacting the movements, and learning to perform with increasing skill. Motor control is achieved by the coordinated activity of myriad neural structures, including the cerebral cortex, basal ganglia, cerebellum, and spinal cord. Neurophysiological recordings, inactivation experiments, and stimulation studies have been used to describe the role of cortex in motor control. Cortical neurons display dynamic activity patterns during movement planning and execution, but their functional contribution to skilled action cannot be assessed from recordings alone (*Lemon, 1993*; *Scott, 2003*; *Evarts, 2011*). Previous inactivation experiments have suggested the cortex tunes—rather than initiates and executes—motor programs to achieve dexterous movement (*Walker and Fulton, 1938*; *Lawrence and Kuypers, 1968*; *Castro, 1972*; *Passingham et al., 1983*; *Martin and Ghez, 1991*; *Whishaw, 2000*; *Fogassi et al., 2001*; *Peters et al., 2014*). The classical perturbation methods used in these studies lack temporal specificity and allow for the emergence of compensatory mechanisms (but see supplement of *Peters et al., 2014*). Electrical stimulation of cortex has been shown to be sufficient to elicit complex movements, but it remains difficult to resolve if evoked behaviors are caused by direct cortical neuron stimulation or antidromic activation of inputs to cortex (*Ferrier, 1873*; *Penfield and Boldrey, 1937*; *Penfield, 1954*; *Gottlieb et al., 1993*; *Graziano et al., 2002*; *Ramanathan et al., 2006*; *Harrison et al., 2012*). Thus the precise role of cortex in skilled movement remains unclear. Here we inhibited cortex with an optogenetic method that minimizes compensation, and selective stimulation of cortical neurons was achieved by the cessation of inhibition.

**eLife digest** Many of the movements that humans and other animals make every day are deceptively complex and only appear easy because of extensive practice. For example, picking up an object involves several steps that must be precisely controlled, including reaching towards the item and holding it using the right amount of pressure to not crush it or drop it. Part of the brain called the motor cortex is thought to be important for learning and controlling these skilled movements, but its exact role in these processes is not clear.

A technique called optogenetics allows the roles of individual parts of the brain to be studied by rapidly altering their activity, whilst minimizing the likelihood that the brain will compensate for these changes. By genetically modifying animals to produce light-sensitive channel proteins in certain brain cells, the activity of particular regions of the brain can be controlled by shining light onto them. Guo et al. have now used optogenetics to control the motor cortex as the mice performed a task they had been trained to do – reaching for and picking up a food pellet.

Suddenly shutting down the motor cortex at the start of a trial prevented the mice from starting the task, and shut down part way through the task caused the front limbs of the mice to freeze in midair. However, only the learned, skilled task was frozen by motor cortex shutdown; mice could still move their limbs normally if the motor cortex was instead shut down during routine movements. When the cortex was reactivated, the mice instantly resumed trying to pick up the food pellet.

Unexpectedly, even during rest periods when there was no food pellet and the mice were just waiting for the experiment to begin, turning the motor cortex off and then back on again suddenly caused the mice to perform the complete grabbing motion. This implies that the cortical activity evoked at the end of inactivation acts to trigger the full movement sequence. This was particularly likely to occur if the animal had been deprived of food before the test or was particularly well trained, but did not depend on the position of the limb.

Overall, Guo et al.'s work opens the question of how the instructions that describe the learned movement are encoded within the motor cortex and its downstream networks. Future studies could also investigate how learning a set of movements affects the structure of cortical neurons and their connections, thus suggesting how these memories are stored.

## Results

To test the role of the cortex for control of skilled motor tasks, we developed a multi-step, cued, prehension task in which head-fixed mice reach for a food pellet, grab it, and bring the pellet to the mouth for consumption (*Figure 1A*, *Video 1*). This prehension behavior was broken down into six sequential epochs (Lift, Hand open, Grab, Supinate, At mouth, and Chew; *Figure 1B and C*) that divide the complex movement into components that involve distinct muscle activations. Unlike previous paradigms, our task employed head fixation to reduce postural variability, allow examination of reaction time, and enable future physiological investigations (*Whishaw et al., 1991*; *Osborne and Dudman, 2014*). All animals that underwent training learned the task. Reaching for the pellet could be achieved in 5–10 sessions and stable performance (median success rate was 74%, n = 13 animals) could be realized in approximately 15 sessions. We used high-speed video and machine-learning-based tracking to semi-automatically convert filmed movements into hand trajectories (see Materials and methods, *Figure 1D and E*, *Video 2*). Stereotypy of head-fixed prehension can be seen in sequential trajectories from a single animal during a behavioral session (*Figure 1E*).

While chronic cortical perturbations degrade prehension performance in freely moving animals, such experiments allow the animal sufficient time to learn behavioral strategies to circumvent affected circuits (*Whishaw et al., 1991*; *Whishaw et al., 1993*). To avoid this confound, we employed rapid onset, within-trial suppression by light-activating channelrhodopsin (ChR2)-expressing inhibitory neurons (expressing the vesicular GABA transporter [*Slc32a1*]) in cortex with pulses of light (*Figure 2*) (*Guo et al., 2014*). This method presumes that activation of *Slc32a1*-COP4*H134R/EYFP-positive neurons selectively suppresses cortex, although other possibilities do exist (for evidence against, see methods). Cortical inhibition was limited in space (*Figure 2—figure supplement 1*) and time by regulating the position and duration of light exposure. To limit compensation, light

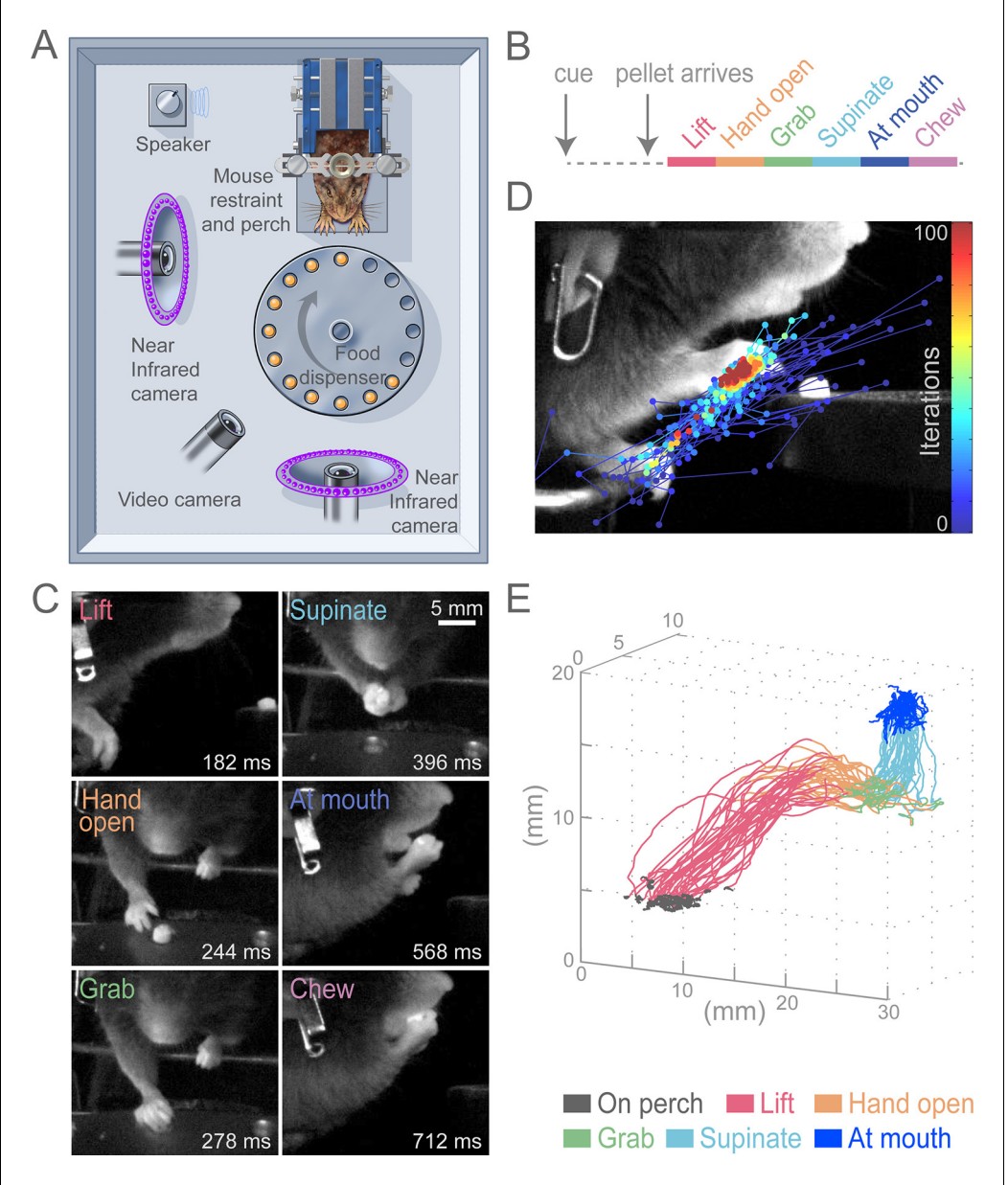

**Figure 1.** Head-fixed, cued, multi-step, prehension task. (A) Schematic of behavioral arena. Mice reached for and grabbed a food pellet presented on a turntable after an auditory cue. High-speed, near-infrared-sensitive cameras captured video of behavior from perpendicular angles. (B) Timeline of prehension components. Prehension behavior consists of the following elements: lift hand from perch, open hand while reaching, grab pellet, supinate hand, elevate hand to mouth, and chew pellet. (C) Example video frames (front or side views) of each prehension component; times are relative to cue. (D) Visualization of iterations of the Cascaded Pose Regression algorithm used to track the forelimb. Each line corresponds to a random initialization (n = 50 initializations) and represents the pose estimates for 100 successive iterations of the tracking algorithm (color indicates iteration). Red dots indicate the distribution of final pose estimates. Our algorithm finds a smooth trajectory over time with high density in this distribution across all frames. (E) Three-dimensional view of stereotyped prehension trajectories (50 trials) of a well-trained animal extracted using the Cascaded Pose Regression algorithm. Behavioral components in the trajectories are color coded as indicated.

was delivered on less than 20% of trials during a behavioral session. We first applied inhibition at cue onset to test the role of cortex for behavior initiation. The only effective area of cortical suppression was centered over contralateral forelimb motor cortex (*Figure 2B–E*, *Video 3*). Because forelimb motor and sensory cortices are overlapping at this resolution in the rodent, we will refer to this region as sensorimotor (SM) cortex (*Ayling et al., 2009*).

Contralateral sensorimotor cortex suppression at the beginning of an individual trial eliminated initiation of the prehension behavior (initiation only occurred in 35 of 459 trials during contralateral sensorimotor cortex inhibition, compared to initiation in 3675 of 3692 control trials, n = 8 animals, *Figure 2A and B*, *Video 3*). Prehension reliably occurred after optogenetic cortical suppression was relinquished (initiation in 436 of 467 trials, n = 8 animals), *Figure 2B*, *Video 3*). Inhibition of other cortical areas did not affect behavioral initiation rates (ipsilateral sensorimotor (initiation in 124 of 124 trials, n = 3 animals, *Figure 2C*, *Video 4*), visual (initiation in 122 of 124 trials, n = 3 animals, *Figure 2D*), and frontal (initiation in 124 of 124 trials, n = 3 animals, *Figure 2E*) cortex). In contrast to prehension, voluntary grooming was not impacted by sensorimotor cortical suppression (11/11 trials, n = 4 animals, *Video 5*). To rule out that initiation failure was due to an inability to process

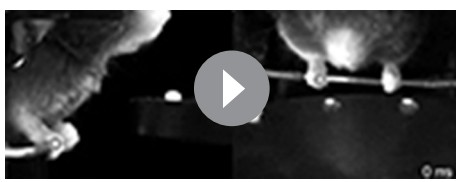

**Video 1.** Example of successful control prehension. Side and front views of head-fixed prehension task. Animal responded to auditory cue and pellet delivery by executing stereotyped components of prehension, acquiring food pellet in single attempt. Hand position (circle) and trajectory (line) as determined by a Cascaded Pose Regression algorithm (color represents behavioral component as indicated). Playback at 50 ms/s.

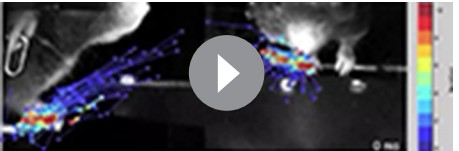

**Video 2.** Visualization of the Cascaded Pose Regression algorithm used to track the forelimb for each frame of a video. In each frame, each line represents 1 of 50 random initializations (in space), with the blue to red color coding representing the progression through the first 10 of 100 iterations of the current frame's pose estimate. Playback at 50 ms/s.

the cue, we activated cortical inhibitory neurons during a cued licking task in which a pellet of food positioned close to the mouth was captured by the tongue; this behavior was not affected by sensorimotor inhibition (*Figure 2—figure supplement 2*, *Videos 6* and *7*). During this cued licking task, animals sometimes moved their forelimb from the perch to their mouth/table (2/3 animals, *Video 6*).

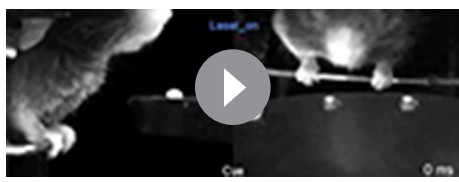

**Video 3.** Optogenetic inhibition of contralateral sensorimotor cortex prevents prehension initiation. Auditory cue and pellet delivery were paired with contralateral sensorimotor cortex inhibition (4 s of light delivery) in *Slc32a1*-COP4*H134R/EYFP mice, preventing mice from initiating prehension. Prehension was initiated at the termination of cortical inhibition. Side and front views of head-fixed mouse. Playback at 100 ms/s.

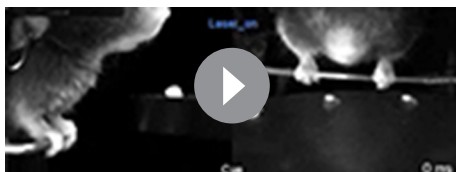

**Video 4.** Optogenetic inhibition of ipsilateral sensorimotor cortex does not affect prehension. Auditory cue and pellet delivery were paired with ipsilateral sensorimotor cortex inhibition (4 s of light delivery) in *Slc32a1*-COP4*H134R/EYFP mice. Mouse performed normal prehension behavior without delay and successfully acquired pellet in single attempt during inhibition of ipsilateral sensorimotor cortex. Side and front views of head-fixed mouse. Playback at 100 ms/s.

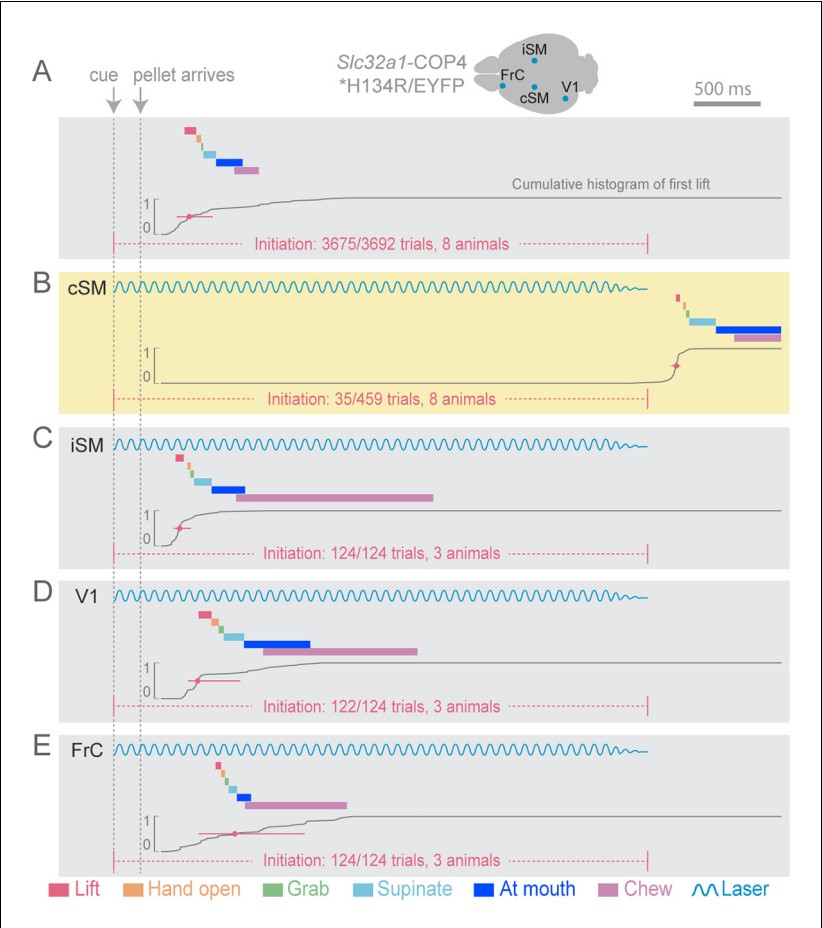

**Figure 2.** Initiation of head-fixed prehension behavior requires contralateral sensorimotor cortex. Optogenetic stimulation of GABAergic inhibitory neurons in *Slc32a1*-COP4*H134R/EYFP mice was used to suppress specific cortical areas, as depicted in the schematic. Laser emission (40 Hz sinusoidal train for 4 s, blue sinusoidal lines) directed through a cleared skull over each cortical area. Depictions of representative ethograms and cumulative histograms for a representative animal. Cumulative histogram is the running total (for all trials) of the occurrence of first list. (**A**) Tone-cued prehension behavior under control conditions with no optogenetic stimulation. (**B–E**) Optogenetic inhibition of cortical regions at trial onset. (**B**) Contralateral sensorimotor cortex (cSM, yellow shading). (**C**) Ipsilateral sensorimotor cortex (iSM). (**D**) Visual cortex (V1). (**E**) Frontal cortex (FrC). Optogenetic inhibition of cSM; but not iSM, V1, or FrC; prevented prehension. Initiation rates over bracketed time period for control and laser trials are listed per condition. Color-coded bars denote behavioral components for (**A–E**).

The following figure supplements are available for figure 2:

**Figure supplement 1.** Quantification of optogenetic laser intensity at different neural depths.

**Figure supplement 2.** Cortical inhibition does not affect cued licking behavior.

While these movements shared some features with the trained prehension behavior, they were not kinematically stereotyped. Contralateral sensorimotor cortical suppression did not prevent these forelimb movements during cued licking (forelimb movement in 44/44 control trials and 20/20 laser trials for two animals, *Video 7*). These results indicate that contralateral sensorimotor cortex activity is necessary for initiation of a learned, goal-directed, skilled prehension behavior.

We next tested the role of cortex during the execution of prehension by activating inhibitory neurons at different times after initiation (*Videos 8–16*). Sensorimotor cortical suppression during the Lift, Hand open, Grab, and Supination components of the movement resulted in a "freezing" of progression towards the target, after a short latency (5/5 animals; *Figure 3A–D*; *Videos 8*,

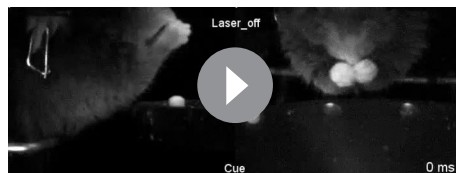

**Video 5.** Optogenetic inhibition of contralateral sensorimotor cortex does not affect forelimb movements during grooming. Auditory cue and pellet delivery delivered during grooming. Contralateral sensorimotor cortex inhibition (2 s of light delivery) in *Slc32a1*-COP4*H134R/EYFP mice did not prevent ongoing grooming. Side and front views of head-fixed mouse. Playback at 100 ms/s.

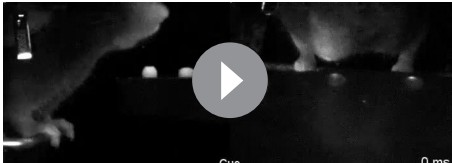

**Video 6.** Cued licking task. Set-up is similar to cued prehension task except food pellet was delivered in close proximity to the tongue. After the cue and delay, the animals used their tongue to acquire the pellet. Forelimbs and hands are used to assist in chewing. Side and front views of head-fixed mouse. Playback at 100 ms/s.

*9*, *10*, *12* and *16*). The latency to freezing (between 72–170 ms for a representative animal) implies that this is the timescale over which the cortex commands specific components of the movement (*Figure 3A4–D4*). Movement during the delay was not simply the result of momentum, since in many cases the movement progressed to the next distinct component before freezing (*Figure 3A3–E4*). Cortical suppression was not effective at impeding hand movements involved in chewing, suggesting these motor programs do not require sensorimotor cortex (*Figure 3E and F*, *Videos 11* and *13–15*), despite involving similar muscle groups. Prolonged suppression of sensorimotor cortex during freezing resulted in a retraction of the limb either to the perch or a characteristic midair position (*Figure 3A2–E2*, *Videos 8*, *9*, *10* and *12*). These results indicate that ongoing contralateral sensorimotor cortex activity is necessary for progressing through the steps of a learned, goal-directed, skilled prehension behavior.

Cortical suppression-induced behavioral freezing was rapidly reversible; immediately upon cessation of the light, animals again reached/grabbed for the pellet ("rebound prehension", 8/8 animals, *Figure 2B*, *Figure 3A1–E1*, *Videos 3*, *8–10*, *12* and *16*). The initiation of rebound prehension had shorter and less variable latencies when compared to control (uninterrupted, tone-cued) prehension (*Figure 2A and B*, *4A*). Once initiated, however, durations of the distinct phases of rebound prehension and the trajectories were not statistically different from control behavior (*Figure 4A–C*; *Figure 4—figure supplement 1*). End-point accuracies of control and rebound prehension attempts were similar (4/5 animals, Wilcoxon–Mann–Whitney rank sum, $p > 0.05$; for 1/5 animal difference was significant but median end-point error was only 0.74 +/– 0.52 mm (median +/– median absolute deviation); *Figure 4D and E*). Interestingly, end-point accuracies were also similar for rebound prehension attempts either beginning from the perch or from a diversity of midair positions (5/5 animals, Wilcoxon–Mann–Whitney rank sum, $p > 0.05$; *Figure 4D and E*). Therefore, motor programs

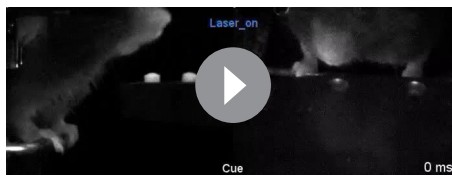

**Video 7.** Optogenetic inhibition of contralateral sensorimotor cortex does not affect cued licking and associated forelimb movements. Auditory cue and pellet delivery was paired with contralateral sensorimotor cortex inhibition (2 s of light delivery) in *Slc32a1*-COP4*H134R/EYFP mice. Lick performance and associated arm movements were not affected by cortical suppression. Side and front views of head-fixed mouse. Playback at 100 ms/s.

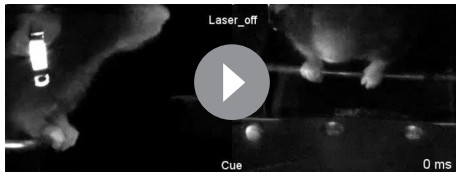

**Video 8.** Prehension progression is blocked by optogenetic inhibition of contralateral sensorimotor cortex during Lift. Contralateral sensorimotor cortex inhibition (2 s of light delivery) in *Slc32a1*-COP4*H134R/EYFP mice occurred during Lift. Progression to the pellet was blocked. Prehension was initiated at the termination of cortical inhibition. Side and front views of head-fixed mouse. Playback at 100 ms/s.

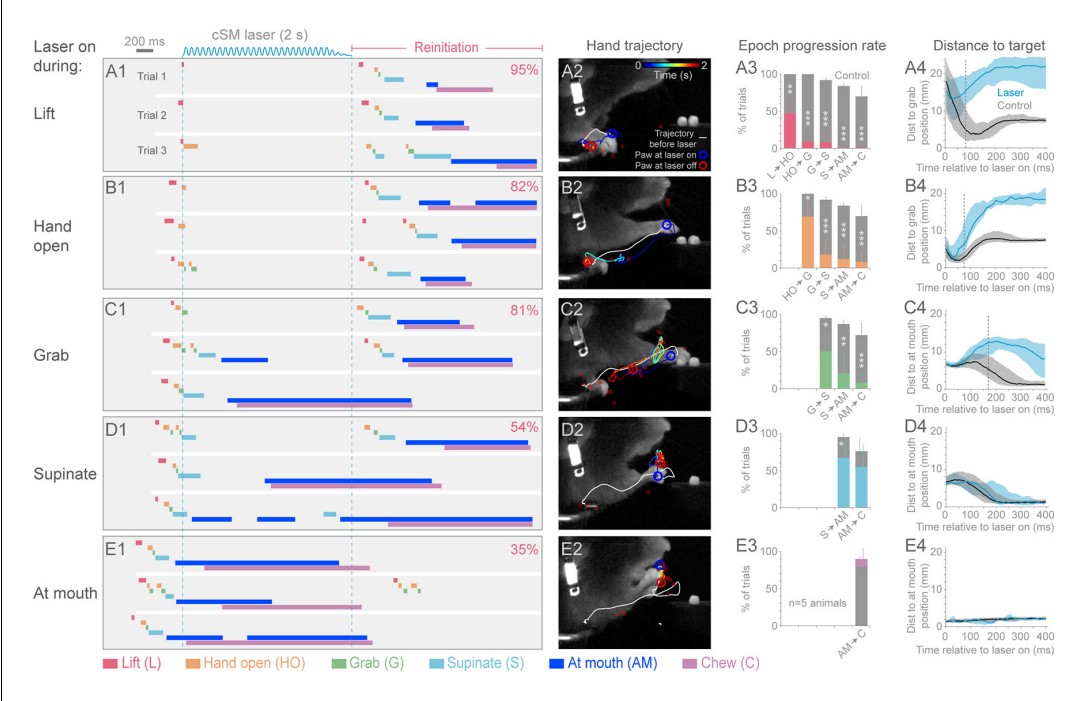

**Figure 3.** Contralateral sensorimotor cortex is required for the ongoing execution of head-fixed prehension behavior. (A1 to E1) Optogenetic activation of contralateral sensorimotor cortical inhibitory neurons of *Slc32a1*-COP4*H134R/EYFP mice during the epochs of tone-cued prehension. Three example trials presented as ethograms for a representative animal. Laser duration marked by sinusoidal and dashed blue lines; behavioral components denoted by color-coded bars (abbreviations indicated). Percentage (top-right, pooled across 5 animals, rate during bracketed time period) of trials where animals reinitiated the behavior upon cessation of cortical inhibition. (A2 to E2) Hand trajectory before and during cortical inhibition of each prehension component. White line labels trajectory from trial start to the onset of inhibition. Colored line represents trajectory during the laser, blue to red color variation represents passing time. Open circles mark hand position at start of inhibition (blue) and end of inhibition (red, small red circles mark laser off position for additional trials). (A3 to E3) Histograms of epoch progression rate, averaged over 5 animals. Control trials are gray and laser trials are the color code of the epoch ongoing at laser onset. For each animal, the number of control trials plotted was equal to the number of laser trials. Control trials were selected based on achievement of at least the epoch ongoing at laser onset. X-axis defines epochs of progression, y-axis represents percent of trials where progression was successful. Progressions where number was not available were removed from the x-axis. Bars are standard deviation. Asterisks indicate statistical significance (t-test with unequal variance, *p<0.05, **p<0.01, ***p<0.001). (A4 to E4) Path deviations during cortical inhibition of a representative animal. Measurement is the median shortest distance between hand position and either the median Grab or At mouth position. Time is in reference to laser onset. Lines represent control trials (black) and trials with optogenetic cortical inhibition (blue, see Materials and methods). Shaded area surrounding the line represents the 25th and 75th percentiles. Vertical dashed line marks the time when the 25th and 75th percentiles no longer overlap.

comparable to natural forms of the behavior follow termination of inhibitory neuron activation in sensorimotor cortex.

Two possibilities could underlie rebound prehension: the animals could make them voluntarily or they could be a direct consequence of the relief of inhibition. An involuntary nature of rebound prehension was supported by its occurrence even on cued trials when the animal had already grabbed the pellet (*Figure 3*, *Videos 10*, *12* and *16*). To further test this, we removed both the pellet and the cue to disengage the animal from the task. During these uncued experiments, the animals remained motionless during contralateral sensorimotor cortical inhibition as expected. Surprisingly, immediately (~250 ms) following light cessation, the animals reached and grabbed at the empty pellet tray (initiation in 192/224 trials, n = 8 animals, *Figure 4F*, *Videos 17*). Rebound execution did not occur in untrained animals (3/3 animals) and was specific to prehension; no other type of movement was generated following light termination (n = 1434 trials, n = 5 animals). If a pellet was in the target position but the animal was not alerted of its presence, cessation of inhibition frequently generated the full reach-grab-eat sequence (*Videos 18*). Taken together, these observations indicate that rebound prehension is not the result of ongoing sensory information or the animals' remembered

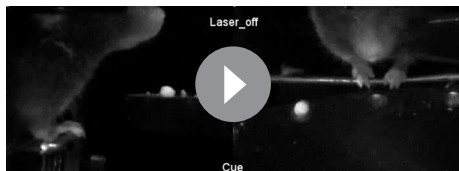

**Video 9.** Prehension progression is blocked by optogenetic inhibition of contralateral sensorimotor cortex during Hand open. Contralateral sensorimotor cortex inhibition (2 s of light delivery) in *Slc32a1-*COP4*H134R/EYFP mice occurred during Hand open. Animal progressed to Grab and then froze. Prehension was initiated at the termination of cortical inhibition. Side and front views of head-fixed mouse. Playback at 100 ms/s.

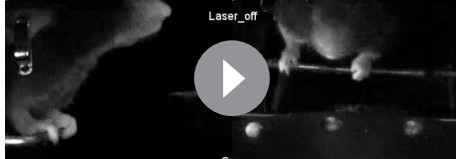

**Video 10.** Prehension progression blocked by optogenetic inhibition of contralateral sensorimotor cortex during Grab. Contralateral sensorimotor cortex inhibition (2 s of light delivery) in *Slc32a1-*COP4*H134R/EYFP mice occurred during Grab. The animal was not able to get the food into the mouth.

Prehension was initiated at the termination of cortical inhibition. Side and front views of head-fixed mouse. Playback at 100 ms/s.

behavioral state. Rather, termination of inhibitory neuron activation in sensorimotor cortex, in combination with plastic changes induced by training, is sufficient to evoke a complex, learned motor act.

Rebound prehension showed dependence on a number of factors. Anatomically, no area outside of sensorimotor cortex contralateral to the utilized limb was effective in evoking the behavior in the no cue/no pellet condition (ipsilateral sensorimotor, 0/68 trials, n = 3 animals, *Figure 4G*; frontal, 0/20 trials, n = 2 animals; visual, 0/21 trials, n = 3 animals). Duration of cortical inhibition was also a critical factor in initiating rebound prehension. At least 2 s (median across animals) of cortical inhibition was required to achieve rebound prehension rates above 90% (n = 4 animals, *Figure 4H*). Longer durations (4 s was longest tested) of cortical inhibition also reduced the latency variability of prehension initiation (n = 2 animals, *Figure 4H*). Interestingly, long durations of photostimulation (2 s or more) were specifically associated with prominent rebound spiking in cortical neurons (*Figure 4—figure supplement 2*). This raises the possibility that rebound spiking in the cortex is responsible for rebound prehension. Rebound prehension itself is also cortically dependent, as reactivation of sensorimotor inhibitory cortical neurons freezes progression of the rebound movement (*Videos 19*, *Figure 4—figure supplement 3*). The presence of food in the mouth decreased rebound prehension rates (18/95 trials exhibited rebound; *Figure 4I*; *Videos 11*, *13–15*). For nearly all of these (93/95) trials, animals were still chewing at light termination. To distinguish whether the reduced rate of rebound prehension was related to motor-program competition (chewing versus prehension) or to motivation (due to consumption of the pellet), we assayed rebound rates for failure trials where animals were chewing without the pellet (29/256 trials, *Figure 4I*). Prehension generally

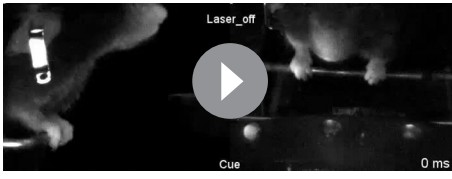

**Video 11.** Prehension progression impeded but not prevented by optogenetic inhibition of contralateral sensorimotor cortex during Grab. Contralateral sensorimotor cortex inhibition (2 s of light delivery) in *Slc32a1-*COP4*H134R/EYFP mice occurred during Grab. Trajectory to the mouth was impeded, but interestingly the unaffected arm was able to facilitate task completion. Prehension was not initiated at the termination of cortical inhibition. Side and front views of head-fixed mouse. Playback at 100 ms/s.

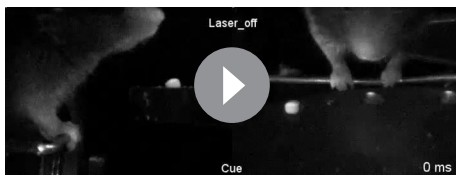

**Video 12.** Optogenetic inhibition of contralateral sensorimotor cortex during Supinate can stop progression to At mouth. Contralateral sensorimotor cortex suppression (2 s of light delivery) in *Slc32a1-*COP4*H134R/EYFP mice occurred during Supinate. The mouse was unable to deliver the pellet into the mouth. Prehension was initiated at the termination of cortical inhibition. Side and front views of head-fixed mouse. Playback at 100 ms/s.

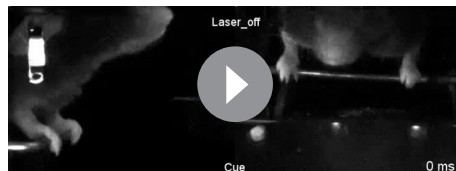

**Video 13.** Optogenetic inhibition of contralateral sensorimotor cortex during Supinate can fail to stop progression to At mouth. Contralateral sensorimotor cortex inhibition (2 s of light delivery) in *Slc32a1*-COP4*H134R/EYFP mice occurred during Supinate. Mouse was able to deliver pellet to mouth. Prehension was not initiated at the termination of cortical inhibition. Side and front views of head-fixed mouse. Playback at 100 ms/s.

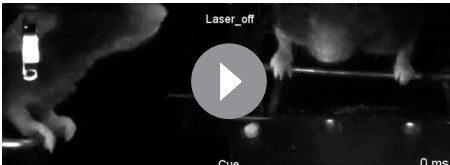

**Video 14.** Prehension is not impeded by optogenetic inhibition of contralateral sensorimotor cortex during At mouth epoch. Contralateral sensorimotor cortex inhibition (2 s of light delivery) in *Slc32a1*-COP4*H134R/EYFP mice occurred during At mouth. Animal was able to deliver pellet into mouth. Prehension was not initiated at the termination of cortical inhibition. Side and front views of head-fixed mouse. Playback at 100 ms/s.

occurred (21/29 trials) if the light was terminated during pelletless chewing. To further test the role of hunger/satiety, we terminated food deprivation; animals on free food for one day exhibited significantly reduced rates of rebound prehension despite still engaging in the control version of the behavior (*Figure 4J*). These results suggest that motivational drive, operating along both short (eating) and long (sated) time scales, gates rebound prehension. Therefore, evocation of the learned behavior depends not only on features of the inhibition but also on the satiety of the animal.

## Discussion

By leveraging the rapid and reversible nature of optogenetic inhibition, we were able to probe the online role of cortex in a learned prehension task. Contrary to the view that cortex is only involved in achieving dexterity (*Kawai et al., 2015*), we found cortex to be necessary for initiating and executing the dexterous and non-dexterous steps of this skilled movement. The necessity of cortex for initiation and execution is consistent with some interpretations of experiments where cortical stimulation delayed or interrupted voluntary movements in primates (*Penfield and Jasper, 1954*; *Day et al., 1989*; *Lemon et al., 1995*; *Churchland and Shenoy, 2007*). Apparent disagreements between our work and that described by Kawai et al. may be due to differences in the behavior and/or the perturbation methodology. The forelimb-lever press task in the study by Kawai et al. requires animals to learn a time interval but not the precise spatial position of an external target. In contrast, the prehension behavior studied here requires the animal to learn the position of the pellet and how to precisely execute a reach to that location. This difference suggests that cortex is not required for interval tasks but rather skilled interactions with objects in the environment. An outlier animal

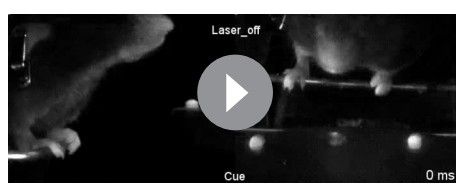

**Video 15.** Chewing is not impeded by optogenetic inhibition of contralateral sensorimotor cortex. Contralateral sensorimotor cortex inhibition (2 s of light delivery) in *Slc32a1*-COP4*H134R/EYFP mice occurred during chewing. Animal continued chewing. Prehension was not initiated at the termination of cortical inhibition. Side and front views of head-fixed mouse. Playback at 100 ms/s.

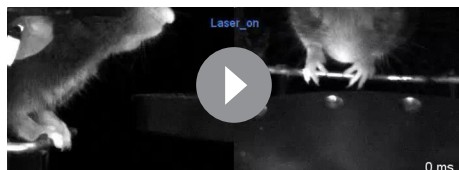

**Video 16.** Relief of brief inhibition of contralateral sensorimotor cortex while animal holding pellet initiates a new prehension attempt. Contralateral sensorimotor cortex inhibition (0.2 s of light delivery) in *Slc32a1*-COP4*H134R/EYFP mice occurred while animal was holding the pellet. Despite this sensory information, the termination of the brief cortical inhibition caused the animal to drop the pellet and initiate a regrabbing at the target position. Side and front views of head-fixed mouse. Playback at 100 ms/s.

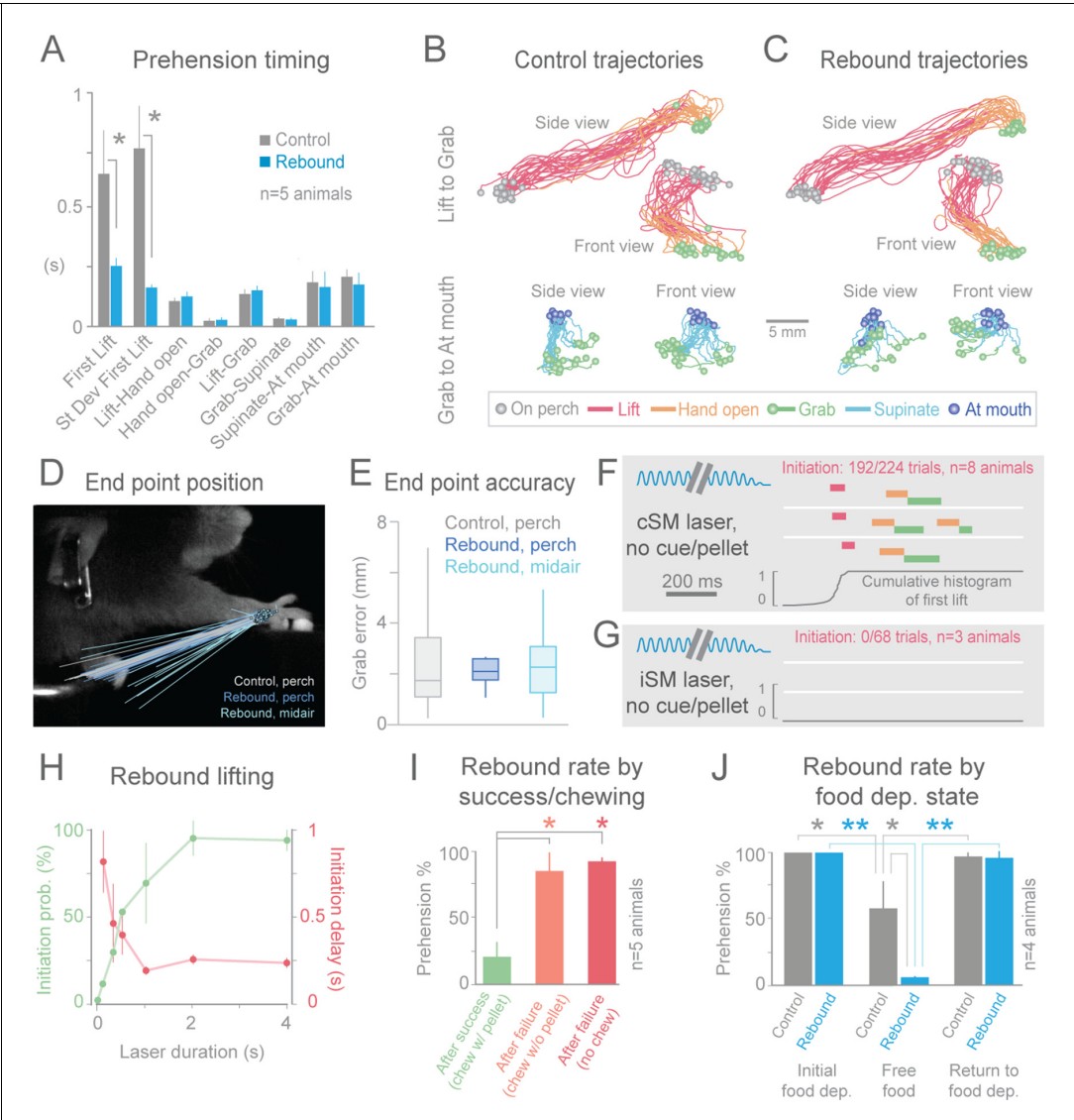

**Figure 4.** Termination of cortical inhibition is sufficient to generate normal prehension behavior. (**A**) Timing differences between control prehension behavior and prehension after laser termination ("rebound prehension"), averaged across 5 *Slc32a1*-COP4*H134R/EYFP animals. Values include First Lift (difference between First Lift and either the cue (control) or laser off (rebound)) and the intervals between sequential behavioral components. Within individual animals, First Lift timing (3/5 animals) and variance (5/5 animals) were also significantly different (Wilcoxon–Mann–Whitney rank sum, p<0.05). (**B** and **C**) Comparison of the trajectories of control (**B**) and rebound prehension (**C**) Trajectories (n = 20 trials) color coded by behavioral component. Top row displays trajectories from Lift to Grab, bottom row represents trajectories from Grab to At mouth. Circles represent on perch (gray), Grab (green), and At mouth (dark blue) hand position. (**D**) Rebound prehension was accurately performed from a diversity of starting positions. Control prehension from the perch (gray, n = 25 trials), rebound prehension from perch (dark blue, n = 25 trials), and rebound prehension from mid-air starting position (light blue, n = 18 trials) for a representative animal. Lines connect starting position of the reach to Grab position. Color-coded circles represent Grab position. (**E**) End-point quantification for control, rebound prehension from perch, and rebound prehension from midair trials for a representative animal. Box plots (whiskers mark minimum and maximum non-outliers) of Grab error: Grab error is defined as the distance between Grab position of an individual trial and median Grab position (control, success trials). No significant difference between control and rebound prehension from the perch or from midair positions (4/5 and 5/5 animals, respectively; Wilcoxon–Mann–Whitney rank sum, p>0.05). (**F**) Extinguishing the laser over contralateral sensorimotor cortex elicited rebound prehension even in the absence of a cue and pellet. (**G**) Laser-off over ipsilateral sensorimotor cortex did not generate rebound prehension. (**F**) and (**G**) Behavioral components denoted by color-coded bars. Blue sinusoidal line marks laser duration. Cumulative histogram of First Lift is shown. (**H**) Effect of laser duration on the likelihood of rebound prehension, initiation latency, and associated jitter in the absence of cue and pellet. Statistically significant (repeated-measures ANOVA, p<0.05) main effect of laser duration on rebound prehension initiation probability, delay, and jitter. (**I**) Rate of rebound prehension as a function of success and chewing status at laser off (average of 5 animals, bars are standard deviation). (**J**) Control (gray) and rebound (blue) prehension rates as a function of food deprivation (dep.) state. Animals

*Figure 4 continued on next page*

*Figure 4 continued*

(n = 4) were taken off food deprivation three times and the rates were averaged. Prehension rates plotted are the means of these averages, bars are standard deviation. Asterisks indicate statistical significance (t-test with unequal variance, *p<0.05, **p<0.001).

The following figure supplements are available for figure 4:

**Figure supplement 1.** Trajectories of control and rebound prehension are similar.

**Figure supplement 2.** Action potential firing rates during and after photostimulation in cortical neurons of awake, non-behaving *Slc32a1*-COP4*H134R/ EYFP mice.

**Figure supplement 3.** Contralateral sensorimotor cortex is required for the ongoing execution of rebound prehension.

described by Kawai et al. provides support for this hypothesis; this animal was unique in that it made precise movements to the lever and showed a significant decrease in performance after cortical perturbation. Additionally, the two experiments also differ in the acuteness of the cortical manipulations; we used rapid onset optogenetic inhibition while Kawai et al. utilized pharmacological and surgical manipulations. The rapidity of optogenetic inhibition may impede compensation mechanisms that ameliorate resulting phenotypes.

We found that termination of inhibitory neuron activation in sensorimotor cortex suffices to drive a full action sequence to the learned target. Previous activation studies of cortex have shown that electrical (*Ferrier, 1873*; *Penfield and Boldrey, 1937*; *Penfield, 1954*; *Gottlieb et al., 1993*; *Graziano et al., 2002*; *Ramanathan et al., 2006*; *Bonazzi et al., 2013*) and optogenetic (*Harrison et al., 2012*) stimulation for behaviorally relevant durations produces motor sequences. Rebound prehension differs from these movements in three ways: rebound prehension is a coordinated sequence of movements (reaching, grabbing, and eating) accurately directed to an external goal regardless of starting position, depends on training, and is gated by motivation (*Ramanathan et al., 2006*; *Harrison et al., 2012*; *Bonazzi et al., 2013*). These differences suggest that termination of inhibition evokes a motor engram specifying a learned goal/end-point of a trained behavior (*Bernstein, 1967*; *Todorov and Jordan, 2002*). Cortical synapses modified during the learning of skilled actions (*Rioult-Pedotti et al., 1998*; *Xu et al., 2009*; *Wang et al., 2011*) may be responsible for the formation of these motor engrams. Alternatively, the engram could be stored elsewhere, and relief of cortical inhibition could trigger or permit its activation in a downstream network. The ability to specifically elicit a post-manipulation transition of cortical state from quiescence to motor engram activation should enable functional characterization of the neural programs responsible for enacting skilled multi-step behavior.

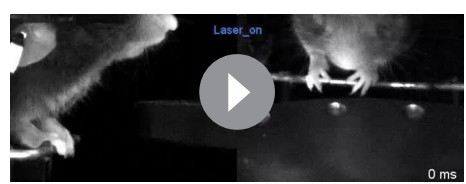

**Video 17.** Even in the absence of a cue or a pellet, cessation of optogenetic inhibition of contralateral sensorimotor cortex evokes prehension. Contralateral sensorimotor cortex inhibition (2 s of light delivery) in *Slc32a1*-COP4*H134R/EYFP mice was not paired with a cue or a pellet. During the laser, the animal remained motionless. Prehension to the remembered target was reliably initiated at the termination of inhibition. Side and front views of head-fixed mouse. Playback at 100 ms/s.

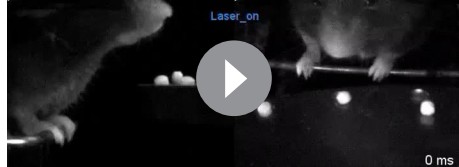

**Video 18.** In the absence of a cue but the presence of a pellet, cessation of optogenetic inhibition of contralateral sensorimotor cortex evokes the full prehension behavior. Contralateral sensorimotor cortex inhibition (1 s of light delivery) in *Slc32a1*-COP4*H134R/EYFP mice was not paired with a cue but a pellet was unexpectedly available.

During cortical suppression, the animal remained motionless. Prehension to the remembered target was reliably initiated at the termination of inhibition; the pellet was grabbed, delivered to the mouth and chewed. Side and front views of head-fixed mouse. Playback at 100 ms/s.

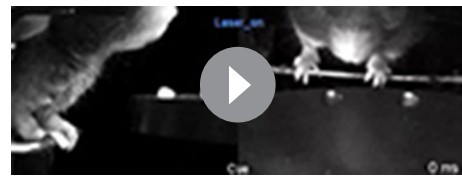

**Video 19.** Optogenetic inhibition of contralateral sensorimotor cortex (3 s of light delivery twice, separated by 320 ms) during Supinate of a post-inhibitory prehension attempt can stop progression to the mouth. Contralateral sensorimotor cortex inhibition in *Slc32a1*-COP4*H134R/EYFP mice occurred during Supinate of a post-inhibitory prehension attempt. Mouse unable to deliver the pellet into the mouth. Prehension was initiated again at the termination of the second cortical inhibition. Side and front views of head-fixed mouse. Playback at 100 ms/s.

## Materials and methods

### Animal care and transgenic mouse lines

Animal procedures were performed in accordance with protocols (Protocol number: 13–99) approved by the Institutional Animal Care and Use Committee (IACUC) of the Janelia Research Campus. Animals were housed on a 12-hr light/dark schedule with *ad libitum* water. Mice undergoing behavioral training were food restricted to 80–90% of original body weight by limiting food intake to 2–3 g/day. Otherwise, mice had ad libitum food. Animals were monitored daily by veterinary staff, and animals recovering from surgery or on food restriction were given a Pain Assessment Score of 1–5, based on IACUC guidelines. Animals with a Pain Assessment Score above 3 were temporarily removed from behavioral testing, given analgesia as determined by the veterinarian, and in some cases their food allotment was increased. Animals were returned to food restriction and behavioral testing after three consecutive days with a Pain Assessment Score below 3. *Slc32a1*-COP4*H134R/EYFP (Stock Number: 014548) were obtained from The Jackson Laboratory (Bar Harbor, ME).

### Cranial window and clear skull and surgeries

Cortical neurons were optogenetically stimulated either through a cranial window or through a cleared skull. Transgenic mice (2–5 months old) were anesthetized with isoflourane (2%) and placed in a stereotactic frame (Kopf Instruments; Tujunga, CA) on a 37°C heating pad. Aseptic technique was used during surgical procedures. The scalp and periosteum over the skull were removed, a layer of UV-curing OptiBond adhesive (Kerr; Orange, CA) (*22*) was applied, and a custom-made headpost was affixed with cement. The clear skull preparation was used for experiments testing the role of different cortical areas. For this preparation, a final smooth layer of clear dental acrylic covered the skull (Jet Repair Acrylic; Lang Dental Manufacturing; Wheeling, IL). Before optogenetic experimentation, the skull was polished using Acrypoints Acrylic Polishing Kit (Pearson Dental; Sylmar, CA) and a thin layer of clear nail polish (Electron Microscopy Services; Hatfield, PA) was applied to reduce glare. Cranial windows (three 170 μm-thick panes of laser-cut glass, 2 mm diameter, glued together with UV adhesive) were placed over sensorimotor cortex.

Following surgery, injections of ketoprophen (5 mg/kg) and buprenorphine (0.1 mg/kg; Henry Schein Animal Health; Melville, NY) were administered subcutaneously. If animals exhibited signs of pain or distress following surgery, additional doses of either ketoprophen or buprenorphine were administered, as directed by veterinary staff. Mice recovered for at least 1 week following surgery and were given ad libitum food and water.

### Behavioral training and data collection

Mice were habituated to head fixation in a custom-built apparatus (adapted from Rodent In Vivo Electrophysiology Targeting System) with forks to secure the head post, adjustable walls to restrain the body, and a perch for the hands (*Figure 1*) (*Osborne and Dudman, 2014*). This apparatus was placed in a light tight, ventilated, soundproof 28-inch cubic behavioral box. A near-infrared-sensitive webcam was used to monitor animals. Mice were initially trained for approximately 30 min per day, until they started licking pellets (10 or 20 mg; Test Diet; St Louis, MO) placed directly below their mouth. Food pellets arrived ~ 200 ms after the start of an auditory tone (5 kHz) by rotating the turntable with a servomotor driven by custom-programmed Arduino software. Mice were initially (1–5 training sessions) trained to retrieve a food pellet by licking and eating the pellet, often using their hand to guide the pellet into their mouth. After cued licking was learned, the turntable was moved progressively further away (over 3–10 sessions) to encourage mice to reach for the pellet after the

cue. After mastering the task (approximately 15 sessions), turntable placement was reproduced daily for each animal by aligning live images to a reference image. Inter-trial intervals ranged from 20 s to 1 min and approximately 50 trials were collected per day. Mice almost always started with hands on perch and trials where animals lifted the hand before the cue were discarded. Mice were trained each day for approximately 60 min until they routinely responded to the auditory cue (within 1 s) and grabbed the pellet. Reprieve of food deprivation was accomplished by administering food ad libitum for one day, prehension rates were tested, and then animals were returned to previous food deprivation conditions.

## Analysis of cued-prehension and cued-licking behavior

Two high-speed, high-resolution monochrome cameras (Point Grey Flea3; 1.3 MP Mono USB3 Vision VITA 1300; Point Grey Research Inc.; Richmond, BC, Canada) with 6–15 mm (f/1.4) lenses (C-Mount; Tokina, Japan) were placed perpendicularly in front and to the right of the animal. A custom-made near-infrared LED light source was mounted on each camera. Cameras were synced to each other and captured at 500 frames/s at a resolution of $352 \times 260$ pixels. Video was recorded using custom-made software developed by the Janelia Research Campus Scientific Computing Department and IO Rodeo (Pasadena, CA). This software controlled and synchronized all facets of the experiment, including auditory cue, turntable rotation, optogenetic lasers, and high-speed cameras. Fiji video editing software was used to label laser onset, termination, and timestamp in the videos.

## Optogenetic activation

To excite ChR2 in cortical inhibitory interneurons through either a cranial window or cleared skulls of *Slc32a1*-COP4*H134R/EYFP mice, we used a 200 µm core, 0.39 numerical aperture fiber (FT200UMT; Thorlabs Inc; Newton, New Jersey) and a fiber-coupled 473 nm laser (LuxX 473–80; Omicron Laserage; Rodgau-Dudenhofen, Germany). Fibers were positioned on or above (1–2 mm) the skull at stereotactic coordinates for different cortical regions (cM1: 0.5 mm anterior to bregma, 1.5 mm lateral of midline; iM1: 0.5 mm anterior to bregma, 1.5 mm lateral of midline; V1: 4.0 mm posterior to bregma, 3.0 mm lateral of midline; FrC: 3.1 mm anterior to bregma, 1.5 mm lateral of midline). 40 Hz sinusoidal stimulation with average power of 1–15 mW at the fiber's end was delivered for 0.1–4 s, and power was reduced to 0 mW over the final 100 ms of stimulation. The approximate diameter of the light spot on the skull/window surface was 400 µm. Laser power of 10–15 mW has been shown to silence spike rates by 90% within 1.5 mm of the center of the laser spot (*Guo et al., 2014*). We found no significant differences between prehension evoked with 2–4 s laser pulses, so they were grouped. To compensate for potential confounds of visible blue light during optogenetic stimulation, the behavioral box was illuminated with a blue LED throughout training and testing. During testing, trials with optogenetic perturbation were interleaved with control trials without laser stimulation (< 20% laser trials, randomly interspersed).

Optogenetic activation of cortical inhibitory neurons and subsequent local cortical suppression most parsimoniously explains the reported phenotypes; however, other mechanisms are possible using the technique employed in this paper. The *Slc32a1*-COP4*H134R/EYFP mouse line used in this study expresses ChR2 in a small set of striatal neurons (*Guo et al., 2014*). This raises a possibility that blue light directed onto the cortical surface activates striatal neurons and this stimulation is responsible for the described phenotypes. We directly measured the transmission of blue light through the cortex into the striatum (*Figure 2—figure supplement 1*). At the depths (+1.8mm) of the striatum, the intensity of the 473 nm light is negligible using our experimental strategy. We also implanted fibers into striatum (FT200UMT; Thorlabs Inc; Newton, New Jersey; coordinates: 1.7 mm lateral of midline; 0.5 mm anterior of bregma; 2.25 mm deep of brain surface) of three trained *Slc32a1*-COP4*H134R/EYFP mice. Delivery of 473 nm light did not halt prehension behavior (prehension initiation in 190/191 laser trials versus 271/271 control trials) in any of these animals. Therefore, it is very unlikely that activation of striatal neurons is responsible for the behavioral phenotypes described in this paper. Another possibility is that long-range GABAergic inputs to the cortex are labeled in the *Slc32a1*-COP4*H134R/EYFP and antidromic activation of these inputs inhibits a non-cortical area that is necessary for prehension (*Saunders et al., 2015*). However, using photostimulation, we were able to produce prehension deficits in mice (n = 2 animals, mouse 1 exhibited prehension initiation in 1/24 laser trials versus 69/72 control trials, mouse 2 exhibited prehension initiation

in 11/21 laser trials versus 67/67 control trials) where viral injection limited ChR2 expression to sensorimotor cortical neurons (AAV 2/1 or 2/7 Synapsin-FLEX-REV-ChR2, 5 sites in sensorimotor (forelimb) cortex, depths of 300 and 600 μm) in Gad2-IRES-Cre (Stock number: 010802; The Jackson Laboratory; Bar Harbor, ME) mice. Finally, long-range inhibitory cortical outputs have been reported more rostral of forelimb motor cortex (*Lee et al., 2014*). Therefore, it is possible that the blue light activates such neurons in motor cortex resulting in inhibition of non-cortical areas that are responsible for the phenotype. However, evidence for GABAergic projection neurons does not exist for areas that produce the phenotypes described in this paper (*Lee et al., 2014*). We also did not detect ChR2-positive axons in the external capsule or corpus callosum of *Slc32a1*-COP4*H134R/EYFP mice (data not shown), further ruling out the possibility that long range cortical inputs and outputs mediate the reported phenotypes.

## Light detection

A custom photodetector *Bittner, et al., 2015* was fabricated by inserting a small plug of 650 nm-emitting quantum dots (Ocean Nanotech; Dunedin, FL) inside the tip of a glass pipette, and sealing the tip in clear epoxy (Henkel; Rocky Hill, CT). Quantum-dot fluorescence excited by laser light was collected by a high-index gel waveguide within the pipette taper and delivered to a 100 μm core optical fiber (0.22 numerical aperture). The collected quantum dot fluorescence was detected using a fiber-coupled QE65000 spectrometer (Ocean Optics; Dunedin, FL).

The photodetector and laser fiber were positioned in parallel directly over a craniotomy above sensorimotor cortex (0.7 mm anterior to bregma, 1.6 mm lateral to midline). For this near-parallel source-detector geometry, the photodetector pipette except the tip was painted black (Liquitex; Cincinnati, OH) to prevent scattering of 473-nm laser light from entering the back of the detector pipette. The photodetector was mounted on a Sutter MP-285 motorized manipulator (Novato, CA) and light intensity measurements were taken at 100 μm increments from the dural surface to 3.0 mm deep. Laser excitation was similar to that used in behavioral experiments: sinusoidal 40 Hz pulses of 1.2 mW peak amplitude for 3 s from a 473 nm fiber-coupled laser.

## Automatic hand tracking

To quantitatively compare tone-cued (control) and laser-evoked (rebound) prehension, we developed a machine-learning-based method to automatically track the mouse's utilized hand. We manually labeled the position of the hand in a small subset of video frames, then used a modified version of Cascaded Pose Regression (*Burgos-Artizzu et al., 2013*) to learn a function that could input a frame and automatically predict the position of the hand in that frame. Cascaded Pose Regression operates directly on a single video frame, and learns a cascade of multiple regressors, which each iteratively bring the target position estimate closer to the ground-truth labels. Our modifications included a different feature-selection method and a multi-pass method for enforcing trajectory smoothness over time. In addition, we developed a new interface for automatically detecting frames in which the tracking might contain errors, and allowing a user to manually fix these.

Using the two-dimensional (2D) trajectories corresponding to the front and side view videos, we reconstructed the three-dimensional trajectories by calibrating the cameras using the Camera Calibration Toolbox for MATLAB (J.-Y. Bouguet, http://www.vision.caltech.edu/bouguetj/calib_doc/ [2004]). A single calibration was initially performed, then fine-tuned per-mouse based on the 2D trajectories.

## Prehension component annotation

Individual prehension component behaviors were manually annotated. *Lift* was defined as the interval from initial separation between hand/perch until the hand was halfway to table. *Hand-open* was defined as the interval from fingers beginning to separate from palm until fully extended. *Grab* was defined as the interval from hand moving downwards as digits close to supination. *Supination* was defined as the interval from the beginning of upward wrist rotation to the hand reached the mouth. *At mouth* was defined as hand in close proximity to mouth. *Chew* was defined as mastication after pellet in mouth. *Chew without pellet* was defined as mastication without successfully retrieving pellet. *Lick* was defined as visible tongue emerging from mouth directed towards pellet until tongue returned to mouth. *Grooming* was defined as using both hands to clean nose, head, or whiskers.

*First Lift* was defined as initial frame of the first Lift after cue. *Lift-Hand open* was defined as the difference between first frames of Hand open and Lift. *Lift-Grab* was defined as the difference between first frames of Grab and Lift. *Hand open-Grab* was defined as the difference between first frames of Grab and Hand open. *Grab-Supinate* was defined as the difference between first frames of Supinate and Grab during the final prehension sequence. *Supinate-At mouth* was defined as the difference between first frames of At mouth and Supinate during the final prehension sequence. *Grab-At mouth* was defined as the difference between first frames of At mouth and Grab during the final prehension sequence.

## Statistical analyses

We used built-in and custom-made scripts within MATLAB (MathWorks; Natick, MA) to perform the following tests: repeated measures ANOVA, Fisher's exact, linear mixed-effects, pairwise t-tests assuming unequal variance, and Wilcoxon–Mann–Whitney rank sum.

## Distance-to-target analysis

In *Figure 3*, we compare the distance to a goal after laser interruption. Per-mouse, we computed the location of the pellet and the mouth as the average position of the hand at the start of Grab and At mouth epochs, respectively, during tone-cued prehension. We separated laser trials by the epoch interrupted, defined as the last component initiated before the laser-on time point. We excluded trials for which the prehension behavior was not impeded. For Lift through Supinate epochs, lack of inhibition was detected as the occurrence of the Chew epoch during the laser-on period. For the At mouth and Chew epochs, lack of an effect was not detectable, thus we included all trials. We constructed a control comparison data set, for example, laser during Lift trials, as follows. For each tone-cued trial, we select the time interval starting $t$ frames after a randomly chosen Lift, where $t$ is the time between the last Lift preceding laser-on and the laser-on time point for a randomly selected laser during Lift trial.

## Nearest-neighbor positional comparison

To measure how similar control and rebound trajectories were, we used a strategy based on nearest-neighbor classifiers. First, we aligned all trajectories in time based on the relative distance traveled in the trajectory. We then asked, for a given distance, how well we could predict whether a trial was a control or rebound trial based on the hand position using a 1-nearest-neighbor classifier. For all mice, the accuracy of this classifier was close to chance. We report the balanced accuracy: the average of the true positive and true negative rates.

## Neuronal recordings and analysis

Extracellular spikes were recorded from *Slc32a1*-COP4\*H134R/EYFP mice (n = 2) using silicon probes (A4 × 8-5 mm-100-200-177; NeuroNexus; Ann Arbor, MI) (*Guo et al., 2014*). Voltage signals (32 channels) were multiplexed, digitized by a PCI6133 board at 312.5 kHz (National Instruments; Austin, TX) at 14 bit, demultiplexed (sampling at 19531.25Hz). Brain movement was minimized by applying silicone gel (3–4680, Dow Corning, Midland, MI) over the craniotomy after the electrode was positioned in the brain. Recordings began several minutes thereafter to allow the brain to settle. Under awake, non-behaving condition, the mice remained idle during different photostimulation conditions. Recordings (n = 2 per animal) were made through the same craniotomy on subsequent days. The extracellular recording traces were band-pass filtered (300–6000 Hz). Events that exceeded an amplitude threshold (4 standard deviations of the background) were subjected to manual spike sorting to extract single units (*Guo et al., 2014*). Eighty-seven single units were recorded under awake, non-behaving conditions. For each unit, spike width was computed as the trough-to-peak interval in the mean spike waveform. Units with spike width <0.35 ms were defined as fast spiking neurons (10/90) and units with spike width >0.45 ms were defined as putative pyramidal neurons (77/90) (*Guo et al., 2014*). Units with intermediate values of spike width (0.35–0.45 ms, 3/133) were excluded from our analyses.

   Effect of photoinhibition on activity was quantified in "average spike rate" across the population. The time course of photoinhibition and rebound activity was computed from averaged peristimulus time histogram (*Figure 4—figure supplement 2*).

## Acknowledgements

We thank V Iyer, A Mittal, and W Dickson for developing acquisition software; J Dudman and J Osborne for designing the RIVET head-restraint apparatus; P Polidoro for developing the Arduino controller; D Flickinger for designing the laser delivery apparatus; G Denisov for guidance with statistics; S Sawtell for constructing infrared LEDs mounted on cameras; J Magee and K Ritola for viruses; Tim Harris and Brian Barbarits for the silicon probe recording system; and G Rubin, J Dudman, A Lee, K Svoboda, J Magee, and M Kabra for helpful conversations and comments on the manuscript. This work was supported by the Howard Hughes Medical Institute.

## Additional information

### Funding

| Funder | Author |
| --- | --- |
| Howard Hughes Medical Institute | Kristin Branson<br>Adam W Hantman |

The funders had no role in study design, data collection and interpretation, or the decision to submit the work for publication.

### Author contributions

J-ZG, Conception and design, Acquisition of data, Analysis and interpretation of data, Drafting or revising the article; ARG, NL, Acquisition of data, Analysis and interpretation of data, Drafting or revising the article; WWG, JZ, Acquisition of data, Analysis and interpretation of data; AL, JR-G, Wrote critical software, Analysis and interpretation of data; JJM, JWP, Acquisition of data, Drafting or revising the article; BDM, Analysis and interpretation of data, Drafting or revising the article; KB, AWH, Conception and design, Analysis and interpretation of data, Drafting or revising the article

### Ethics

Animal experimentation: Animal procedures were performed in accordance with protocols (13-99) approved by the Institutional Animal Care and Use Committee (IACUC) of the Janelia Research Campus.

## Additional files

### Supplementary files

• Source code 1. Code for the analysis described in the article.

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
