## [Decision Letter]

Thank you for submitting your work entitled "Cortex commands the performance of skilled movement" for peer review at *eLife*. Your submission has been favorably evaluated by Timothy Behrens (Senior editor), a Reviewing editor, and three reviewers.

The reviewers have discussed the reviews with one another and the Reviewing editor has drafted this decision to help you prepare a revised submission.

Summary:

Guo et al. explored the role of motor cortex in the execution of a skilled forelimb reaching task. To address this, the authors adopted a prehension task, widely used as a motor skill learning paradigm in rodents, to head-fixed mice. Combining this task with optogenetic inactivation, which allows inactivation with high temporal resolution, the authors show that mid-movement inactivation is still effective in pausing the execution of learned prehension behavior. The authors also make the surprising observation that the cessation of cortical inactivation often coincides with a short-latency, low-jitter execution of the learned movement. This phenomenon is only seen in learned, motivated animals but does not require the presence of actual food pellets. The authors interpret these results as rebound excitation following inactivation being 'sufficient' to activate the motor engram.

The role of cortex in motor learning and execution of learned movements has been the subject of many studies, but is still hotly debatable. Previous studies did not distinguish, for example, whether cortex is necessary just for the initiation of a motor sequence or whether it is necessary for the execution throughout. This study is particularly timely given the controversy raised by a recent study (Kawai et al., Neuron 2015) suggesting that motor cortex in rodents is not necessary for the execution of movements, and is just required for learning. The previous study used mostly lesions and a lever pressing task. In contrast, the present study uses optogenetic inactivation, providing better temporal resolution (less compensation) and a task where more skilled prehension is required. To summarise, the present study provides compelling evidence that motor cortex is necessary – at least when performing this particular task – for movement execution even after the initiation of movement. This is novel and important.

Essential revisions:

1) The cessation-induced prehension behavior is an intriguing observation that is not simple to interpret. One extreme scenario which may be implied by the authors is that cortex contains all the instructive information about the movement sequence that is activated upon cessation. However, many other explanations are also possible, including that the motor sequence information is entirely subcortical and cortical activity simply provides non-specific excitation sufficient to elicit the subcortical activity. We encourage the authors to make a greater effort to discuss these and other possibilities.

2) An important mechanistic question is how inactivation translates to cortical activity. The authors assume rebound excitation, serving as motor cortex specific stimulation. First, it would be helpful to show with a few recordings what kind of activity, if any, is evoked at inactivation cessation. Does it indeed result in reliable low-jitter spiking of most motor cortex neurons? Also, it is unclear if this is in any way different from microstimulation. The authors claim that inactivation cessation achieves area-specific activation, while microstimulation can activate fibers of passage. Area-specific stimulation can be achieved by optical stimulation while restricting the opsin expression to a specific area. It would be interesting to know if this method would reproduce the prehension behavior, or if inactivation cessation is special. The latter possibility would suggest that inactivation is important in setting the system in a certain state that is more likely to produce the learned movement upon stimulation. It would also be helpful to know if an independent method of optogenetic inhibition targeted to principal cells results in rebound prehension (presumably not). While we know that asking for extra data collection here is somewhat counter to *eLife's* policy, it seems particularly pertinent to the major claim of the study to get at least one of these extra datasets, or preferably both.

3) The authors should be more careful in citing previous literature. Most problematically, they claim that 4 of the characteristics of the cessation-induced behaviors are different from those evoked by cortical stimulation. The first two characteristics, namely that it is a coordinated sequence of movements and that it is directed to a goal regardless of the starting position, are actually the very characteristics seen in at least one of the studies they cite (Graziano). Graziano's long stimulation resulted in, for example, the monkeys bringing their arm and hand to their mouth, regardless of the starting position, coinciding with the opening of the mouth. Another example is that, despite their contention that previous studies used temporally non-specific inactivation, Peters et al. used optogenetic inactivation, much the same way used in the current study. Furthermore, despite the contention that their study is contrary to the view proposed by Kawai et al., the current skilled movement being studied would be considered 'dexterous' in their definition and so there is not a contradiction between these previous studies and the present work; further and more balanced discussion of why this report comes to a fundamentally different conclusion to that of Kawai et al. (2015) would therefore be useful. Finally, previous work showing that the effects of transcranial magnetic stimulation of M1 in humans, which also interrupts actions which are then completed several 100 ms later, might be worth citing (e.g. Day et al. 1989, Brain; Lemon et al. 1995 J. Neurosci. Figure 8).

4) Please add more information about the reach and grasp task in the mouse as a model for skilled movement. How long did it take to train the mice used? How many of the mice selected were able to learn the task? What was the average success rate of fully trained mice on the task?

5) Why was the 'chew' phase so extended by SM inhibition (Figure 2)?

6) Figure 2—figure supplement 1: Would you not expect the trained licking task to be affected by (face/jaw/tongue) SM inhibition? How much do the authors know about the cortical extent of spread of the induced inhibition?

7) Why is there no freezing of subsequent action phases in Video 11 (in others such Video 16, the action is stopped and the pellet dropped)?

8) If possible, a slightly higher N of animals would be advisable (at least in some of the experiments: e.g. currently 5/5 in the best cases and 3/3 in some).

[Editors' note: further revisions were requested prior to acceptance, as described below.]

Thank you for resubmitting your work entitled "Cortex commands the performance of skilled movement" for further consideration at *eLife*. Your revised article has been favorably evaluated by Timothy Behrens (Senior editor), a Reviewing editor, and two reviewers. The manuscript has been improved but there are some remaining issues that need to be addressed before acceptance, as outlined below:

The authors' efforts to characterize further their manipulations and discuss some controversial interpretations have strengthened the study, as have the in vivo recordings. However, it is unclear why the authors did not do this during behavior given that their task is head-fixed; and also how pyramidal neurons were separated from fast-spiking neurons. The authors should try to explain their cell classification better, and be more cautious if this classification relied heavily on inhibition and excitation of activity. The authors should also add the data from the additional 3 animals to the main group analyses.

Another remaining concern is about the description of the training procedure. The authors added a small amount of detail to the revised manuscript in response to the initial review. However, they should provide much more detail to make the results reproducible by other labs. Specifically, the methods on behavioral training describe several shaping steps. The authors should provide the specific duration, with the range, for each step.

Finally, since the authors have elected not to do some of the requested experiments (direct excitation of pyramidal cells to analyze the effects of rebound excitation, and/or direct inhibition of pyramidal neurons to see that it does not happen), they should tone down the description of the rebound effects to be more speculative, and focus more on the deficits caused by inhibition. We suggest that they also provide more discussion about different potential mechanisms for the effects they observe.

---

## [Author Response]

*Essential revisions:*

*1) The cessation-induced prehension behavior is an intriguing observation that is not simple to interpret. One extreme scenario which may be implied by the authors is that cortex contains all the instructive information about the movement sequence that is activated upon cessation. However, many other explanations are also possible, including that the motor sequence information is entirely subcortical and cortical activity simply provides non-specific excitation sufficient to elicit the subcortical activity. We encourage the authors to make a greater effort to discuss these and other possibilities.*

We agree that many mechanisms may underlie rebound prehension. We did not intend to suggest that the motor engram was entirely contained within the cortex and we have changed the text accordingly.

“The engram need not be stored entirely within the cortex; rebound cortical activity may be sufficient to trigger or open a gate for an engram stored in a downstream network.”

We have not included other more exotic explanations, for example that driving cortical inhibitory neurons creates an active stop signal, but of course these possibilities do exist.

*2) An important mechanistic question is how inactivation translates to cortical activity. The authors assume rebound excitation, serving as motor cortex specific stimulation. First, it would be helpful to show with a few recordings what kind of activity, if any, is evoked at inactivation cessation. Does it indeed result in reliable low-jitter spiking of most motor cortex neurons?*

Reliable and low jitter spiking at the termination of photostimulation in *Slc32a1-COP4*H134R/EYFP* mice was reported in Guo et al. (2014, Neuron). However these recordings were not performed in cortical areas relevant to the work in this manuscript. Thus, to address this critique, we have collected recordings from sensorimotor cortex and have included this data as a supplemental figure. Reliable spiking in putative pyramidal and fast-spiking neurons) occurs at the termination of photostimulation in *Slc32a1-COP4*H134R/EYFP* mice. Interestingly, prominent rebound spiking specifically occurred under the stimulation parameters that were most effective in generating rebound prehension (Figure 4—figure supplement 2).

Also, it is unclear if this is in any way different from microstimulation. The authors claim that inactivation cessation achieves area-specific activation, while microstimulation can activate fibers of passage. Area-specific stimulation can be achieved by optical stimulation while restricting the opsin expression to a specific area. It would be interesting to know if this method would reproduce the prehension behavior, or if inactivation cessation is special. The latter possibility would suggest that inactivation is important in setting the system in a certain state that is more likely to produce the learned movement upon stimulation.

We agree this is an interesting question. We are currently performing these experiments as well as studying the differences in cortical activity after microstimulation and rebound prehension. This will be the subject of a future manuscript. For the purposes of the present manuscript, because inhibitory neurons do not project, our manipulation is at least as locally restricted as prior methods, and arguably more than microstimulation. For this reason, we assert that this manipulation advances the degree to which the locality of the effect can be inferred.

It would also be helpful to know if an independent method of optogenetic inhibition targeted to principal cells results in rebound prehension (presumably not). While we know that asking for extra data collection here is somewhat counter to eLife's policy, it seems particularly pertinent to the major claim of the study to get at least one of these extra datasets, or preferably both.

We have attempted to inhibit Layer 5 pyramidal neurons using ArchT and Halorhodopsin. We have yet to find an effective strategy for achieving robust and reliable inhibition with these methods.

*3) The authors should be more careful in citing previous literature. Most problematically, they claim that 4 of the characteristics of the cessation-induced behaviors are different from those evoked by cortical stimulation. The first two characteristics, namely that it is a coordinated sequence of movements and that it is directed to a goal regardless of the starting position, are actually the very characteristics seen in at least one of the studies they cite (Graziano). Graziano's long stimulation resulted in, for example, the monkeys bringing their arm and hand to their mouth, regardless of the starting position, coinciding with the opening of the mouth.*

We have combined the first two items in the list. While Graziano did show multi-joint egocentric movements, to our knowledge, he did not report coordinated sequence of movements to an allocentric goal.

*Another example is that, despite their contention that previous studies used temporally non-specific inactivation, Peters et al. used optogenetic inactivation, much the same way used in the current study.*

We agree that in addition to the muscimol experiments we were referring to, optogenetic cortical inhibition was included in the supplement of Peters et al. Although their optogenetic perturbation had coarser temporal resolution, we have now included this in the text to provide a more complete treatment of the relevant literature.

*Furthermore, despite the contention that their study is contrary to the view proposed by Kawai et al., the current skilled movement being studied would be considered 'dexterous' in their definition and so there is not a contradiction between these previous studies and the present work; further and more balanced discussion of why this report comes to a fundamentally different conclusion to that of Kawai et al. (2015) would therefore be useful.*

We have added a new Discussion paragraph that directly compares our study with that of Kawai and colleagues. Given this extra space, we are able to more precisely express the similarities and differences.

*Finally, previous work showing that the effects of transcranial magnetic stimulation of M1 in humans, which also interrupts actions which are then completed several 100 ms later, might be worth citing (e.g. Day et al. 1989, Brain; Lemon et al. 1995 J. Neurosci. Figure 8).*

We include these and a few related references in the text.

4) Please add more information about the reach and grasp task in the mouse as a model for skilled movement. How long did it take to train the mice used? How many of the mice selected were able to learn the task? What was the average success rate of fully trained mice on the task?

Further description of animal performance is included in the first paragraph of the Results section.

*5) Why was the 'chew' phase so extended by SM inhibition (Figure 2)?*

The example trials in Figure 2 do show differences in chew duration, however such differences were not significantly different when comparing the complete datasets of control and photoinhibition trials.

*6) Figure 2—figure supplement 1: Would you not expect the trained licking task to be affected by (face/jaw/tongue) SM inhibition? How much do the authors know about the cortical extent of spread of the induced inhibition?*

We were a bit surprised by the inability to stop the lick task. We attempted to directly photoinhibit the “lick” cortical areas and we were still unable to prevent licking. Perhaps our lick task is not cortically dependent or photoinhibition did not cover a large enough area. For example bilateral inhibition may be required to prevent licking. We have included some discussion of this matter in Figure 2—figure supplement 2.

*7) Why is there no freezing of subsequent action phases in Video 11 (in others such Video 16, the action is stopped and the pellet dropped)?*

The animal actually does freeze in Video 11. We included this video because the animal engages in a very interesting behavior after freezing. The animal uses its unaffected limb to pull the frozen limb to the mouth. This interesting behavior was discussed in the legend, but as a result of this critique, we have now given this behavior a bit more emphasis in this version of the manuscript.

*8) If possible, a slightly higher N of animals would be advisable (at least in some of the experiments: e.g. currently 5/5 in the best cases and 3/3 in some).*

We have included 3 additional animals. These animals are subjects in a follow up study and trial analysis is still ongoing for these animals. Each of these animals shows cessation of movement during the inhibition and shows rebound prehension following photostimulation.

[Editors' note: further revisions were requested prior to acceptance, as described below.]

*The authors' efforts to characterize further their manipulations and discuss some controversial interpretations have strengthened the study, as have the in vivo recordings. However, it is unclear why the authors did not do this during behavior given that their task is head-fixed;*

We agree that performing recordings (both electrophysiology and imaging) in trained animals is a very interesting direction of study. This is currently a major effort in the lab, but requires many new trained animals and detailed analysis. This data will be the subject of a future manuscript.

*And also how pyramidal neurons were separated from fast-spiking neurons. The authors should try to explain their cell classification better, and be more cautious if this classification relied heavily on inhibition and excitation of activity.*

For the last resubmission we added the following text, which clarifies that we classified based on spike width rather than opto-tagging.

“For each unit, spike width was computed as the trough to peak interval in the mean spike waveform. Units with spike width <0.35 ms were defined as fast spiking neurons (10/90) and units with spike width >0.45 ms as putative pyramidal neurons (77/90). Units with intermediate values of spike width (0.35 - 0.45 ms, 3/133) were excluded from our analyses.”

This analysis thus does not rely on opto-tagging approaches. However, as expected on average, putative fast spiking neurons increased their firing during light exposure and putative pyramidal neurons showed a depression of activity (see Figure 4—figure supplement 2). For clarity, we also described the classification scheme in the legend for Figure 4—figure supplement 2.

*The authors should also add the data from the additional 3 animals to the main group analyses.*

We have now added this data.

*Another remaining concern is about the description of the training procedure. The authors added a small amount of detail to the revised manuscript in response to the initial review. However, they should provide much more detail to make the results reproducible by other labs. Specifically, the methods on behavioral training describe several shaping steps. The authors should provide the specific duration, with the range, for each step.*

We added information about the range of sessions for each shaping step.

*Finally, since the authors have elected not to do some of the requested experiments (direct excitation of pyramidal cells to analyze the effects of rebound excitation, and/or direct inhibition of pyramidal neurons to see that it does not happen), they should tone down the description of the rebound effects to be more speculative, and focus more on the deficits caused by inhibition.*

We have changed conclusion sentences throughout the Results sections to be more direct. We have also toned down the following sentence in the Discussion to make the description of the rebound effects to be more speculative.

“The engram need not be stored entirely within the cortex; rebound cortical activity may be sufficient to trigger or open a gate for an engram stored in a downstream network.”

To

“Alternatively, the engram could be stored elsewhere, and relief of cortical inhibition could trigger or permit its activation in a downstream network.”

*We suggest that they also provide more discussion about different potential mechanisms for the effects they observe.*

The above change to the Discussion section directly suggests a different potential mechanism for our observed effects, namely that cortex may be permissive of downstream engrams. We also refer (in the main text) to the Methods section that examines other possible mechanisms using this optogenetic technique.